# Automating Website Registration for Studying GDPR Compliance

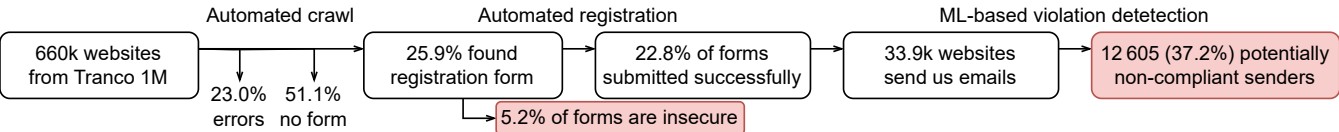

Figure 1: Overview of steps of our study and results.

## ABSTRACT

Investigating how websites use sensitive user data is an active research area. However, research based on automated measurements has been limited to those websites that do not require user authentication. To overcome this limitation, we developed a crawler that automates website registrations and newsletter subscriptions and detects both security and privacy threats at scale.

We demonstrate our crawler's capabilities by running it on 660k websites. We use this to identify security and privacy threats and to contextualize them within the laws of the European Union, namely the General Data Protection Regulation and ePrivacy Directive. Our methods detect private data collection over insecure HTTP connections and websites sending emails with user-provided passwords. We are also the first to apply machine learning to web forms, assessing violations of marketing consent collection requirements. Overall, we find that 37.2% of websites send marketing emails without proper user consent, which is mostly caused by websites sending first a marketing email right after the subscription. Additionally, 1.8% of websites share users' email addresses with third parties without a transparent disclosure.

## CCS CONCEPTS

• **Security and privacy** → **Privacy protections**; **Web application security**; • **Applied computing** → *Law*; • **Information systems** → Web mining.

## KEYWORDS

Crawling, Registration, Consent, GDPR, ePrivacy Directive, Compliance

**ACM Reference Format:**
Anonymous Author(s). 2018. Automating Website Registration for Studying GDPR Compliance. In *Proceedings of Make sure to enter the correct conference title from your rights confirmation emai (Conference acronym 'XX)*. ACM, New York, NY, USA, 11 pages. https://doi.org/XXXXXXX.XXXXXXX

## 1 INTRODUCTION

Since the Internet's beginnings, users have been exposed to security and privacy abuses [8, 28]. Over the past decades, the advertising industry has been in on the game, employing tracking technologies [35] to gather user data. Many of these abuses are financially motivated since users' behavioral data has economic value, e.g., for targeted advertising.

To protect individuals, the European Union (*EU*) has enacted several laws regulating online data collection. The ePrivacy Directive mandates that the sending of marketing emails requires prior consent of the recipient. This consent is currently defined in the General Data Privacy Regulation (*GDPR*). Both of these laws have further requirements that pertain to the processing of personal data, such as following secure communication practices.

Even though data protection authorities can impose heavy fines for GDPR violations, the majority of studies analyzing EU websites' compliance find significant levels of non-compliance. For example, Libert [24] and Englehardt et al. [13] demonstrated that most websites track users through cookies or fingerprinting, respectively. Matte et al. [27] showed that 5% of websites ignore users' cookie consent, and Linden et al. [25] found that almost half of websites' privacy policies violate GDPR requirements.

These studies have focused solely on websites' landing pages. Urban et al. [33] showed that browsing random pages beyond a site's landing page increased the incidence of privacy-invasive practices by 36%. However, their study was also limited to unauthenticated sections of websites, a limitation that has been addressed by only a few researchers. Englehardt et al. [12] and Mathur et al. [26] studied email privacy by signing up for US e-commerce and political campaign newsletters, observing address sharing to third parties and email tracking. Jonker et al. [18] utilized a public credential database to log in to websites. However, their work was limited to websites with available credentials in that database. Only the study by Drakonakis et al. [11] addressed the registration process in general, but it was successful on just 1.6% of the Alexa top 1M websites, finding half of websites using insecure cookies.

*Our work.* We present a crawler that achieves a significantly higher registration and newsletter sign-up rate than previous work; in particular, it allows for the analysis of those parts of websites that require prior user authentication, which have been understudied. Utilizing this infrastructure, we examine the compliance of websites with security and privacy requirements for the registration process and analyze the emails received from these websites. The crawl process and associated statistics are depicted in Fig. 1.

To examine websites' compliance, we trained machine learning (*ML*) models on datasets from Kubicek et al. [21] predicting the legal properties of forms and the received emails. By processing the form's legal properties in decision trees, we can detect various kinds of potential violations of consent to marketing emails. We are able to identify instances where consent is likely not active, free, specific, or given at all, thereby violating GDPR requirements.

We evaluate both crawler and violation detection on a crawl of 660k websites, registering or signing-up for newsletter in 5.9% of them. Using an ML classification of email types, we evaluate the verification process of email address control, known as *double-opt-in*, finding 59.8% of websites that fail to follow this process. Since we generate a unique email address for each registered website, we discovered that in 14.5% of the cases we received emails from domains other than the domain where we registered. We develop methods to evaluate the transparency of their disclosure practices, finding 1.8% of websites with undeclared or hidden senders.

*Contributions.* We make the following key contributions. (1) We develop a crawler that achieves more than double the rate of registration and newsletter sign-up than prior work. Our crawler enables the automated analysis of those parts of websites that require prior user authentication, enabling privacy and security studies at scale that were previously not possible.[1] (2) We automate the detection of privacy and security violations using ML models that allow the fully self-contained processing of crawled registration forms and received emails. (3) We present new results on how tens of thousands of websites potentially violate GDPR consent requirements in the user registration process. Namely 37.2%, which is 12 605, of websites send marketing emails despite insufficient consent. This demonstrates the usefulness of our crawler in analyzing the security and privacy of the registration process.

## 2 LEGAL BACKGROUND

During the registration process, users provide personal information to websites, including their names, passwords, telephone numbers, and email addresses. Within the EU, the collection and processing of such information is regulated by the ePrivacy Directive and the General Data Protection Regulation (*GDPR*). The ePrivacy Directive regulates electronic communication, mandating prior consent (an *opt-in* regime, unlike in the US) for sending marketing emails.

The GDPR defines in Articles 4(11), 7, and Recital 32 the requirements for obtaining consent: it must be freely given, specific, informed, and unambiguous. For example, valid consent is considered to be given when users actively mark a checkbox that explicitly asks for consent to receive marketing emails, accompanied by a clear explanation of what this means. For forms exclusively dedicated to newsletter subscriptions, where the purpose of receiving marketing emails is implicit in the form's wording, the inclusion of a checkbox becomes redundant. Nevertheless, in all cases, websites should first send an activation email to verify the user's possession of the registered address through a *double-opt-in* procedure.

Furthermore, Articles 25 and 32 of the GDPR outline the obligations to follow best practices, namely to implement secure and

---

[1]Our crawler is not publicly available as it can be misused for the Bomb attack studied by Schneider [31]. However, interested researchers can request access using this form (redacted for review).

private data processing by design and by default. This requirement aims to prevent data breaches involving email addresses or passwords, which have led to significant fines [3, 17]. Collecting private data through forms on insecure websites using HTTP, or transmitting user-provided passwords via unencrypted emails may therefore violate the requirement of following best security practices.

## 3 CRAWLING INFRASTRUCTURE

We developed an infrastructure for crawling websites and automating user registration. For each website where the crawler registers, we provide a unique email address for a (simulated) user. Our infrastructure then analyzes the received emails to evaluate how the website uses the user's email address.

### 3.1 Crawler

Websites vary significantly in both their appearance and implementation, primarily due to the flexibility of JavaScript and CSS. Since all registration options must adhere to the same laws regardless of the technologies used, we focus on registration using an email address. We therefore do not attempt to register using single sign-on, which was covered by other compliance studies [10].

Below we discuss the crawling steps. First, the crawler navigates through website to find pages containing a registration form, which it then fills out and submits. Afterwards, it checks the registration state and finishes the double-opt-in when it is requested by email.

*3.1.1 Implementation.* To simulate users' browsing patterns, our crawler utilizes a real browser orchestrated by Selenium. Since existing frameworks such as OpenWPM [13] or webXray [24] are not designed for the complex crawling that our task demands, we do not use them. To represent the majority of web users, we crawl websites using Chrome, but Firefox is supported as well.

To maximize the chances of successfully loading websites, we employ several techniques to evade bot detection, which we describe in Appendix A.1. We have tested that our crawler is not flagged by any major Content Delivery Network (*CDN*), including Cloudflare, Fastly, Amazon CloudFront, and Akamai.

Our crawler successfully loads 90.6% of websites, as opposed to 70% without bot evasion techniques. In comparison, Le Pochat et al. [22], successfully crawled 85% from URLs of a similar list (the intersection of the Tranco and Chrome UX report lists). Their crawler did not actively evade bot detection. We suspect that many of the websites that they report as successfully loaded actually flagged their crawler as a bot and presented a simple warning page.

*3.1.2 Navigation.* After loading each website with a fresh cache, the crawler determines the page's language using the polyglot Python package. If the language detection fails, we rely on the <html> tag. If English is not the detected language, the crawler tries to switch to the English version, if one exists. We keep browsing the website regardless of the switch to English since we support the majority of European languages (see Appendix A.2).

*Keyword matching.* The detection of a link or button to change the language is based on matching keywords in the visible text, the 'alt' attribute of  tag, or the URL. We curated phrases for determining the purpose of page elements, such as a privacy policy link or marketing consent checkbox. Native speakers translated

these phrases to all the supported languages. The curation was guided empirically by example websites. The matching procedure works as follows. First, we remove stop words from both the website and the keyword phrase. Then we lemmatize both texts, using the SpaCy [16] or lemmagen3 [19] lemmatizers, depending on the language support. Next, we map characters with accents or Cyrillic to lowercase ASCII counterparts. Finally, the processed keywords and phrases are matched. This keyword matching approach is also used for other navigation aspects, which are described below.

*Navigating webpages.* Our crawler uses a priority queue to determine the order of visiting pages of the site. The priority represents the likelihood that a given link leads to a registration or a newsletter form. We order the link categories starting with the highest priority as follows: the registration page, login page, privacy policy and terms and conditions, and others. Links within a category are ordered by their matching score. The "other" links are selected randomly, preventing the crawler from getting stuck by, e.g., age walls on adult websites. The privacy policy and terms are collected after registration; they are relevant for our legal evaluation.

The crawler is restricted to visiting at most twenty pages and the registration page is typically reachable within the first five pages. We allow the crawler to navigate beyond the original TLD+1 domain,[2] but only for a single step, i.e., links found on external domains are not considered for subsequent crawling. This allows registration on an affiliated website directly accessible from the original site. However, it restricts the crawler from navigating away from the original site and identifying unrelated registration forms. Moreover, the keyword-matching algorithm penalizes external domains.

*Page content classification.* When we load a page, we classify it according to the presence and thereby type of a <form> tag. We apply the decision tree depicted by Fig. 6 to classify the form as `registration`, `login`, `subscription`, `contact`, `search`, or `other`. We evaluated this procedure on a manually annotated dataset collected from 1000 randomly selected English websites from Tranco 1M,[3] containing 426 forms. There were 12 `contact`, 32 `login`, 139 `subscription`, 163 `registration`, and 80 `other` forms. Procedure from Fig. 6 detected 74% of the `registration` forms and 94% of the `subscription` forms, yielding an overall accuracy of 82%.

3.1.3 *Form interaction.* Once we detect a `registration` form, or a `subscription` form when no `registration` form is found, we interact with it. We first extract the entire subtree of the <form> tag, which we process using the Beautiful Soup library. We use a similar keyword-matching method as in Section 3.1.2 to detect the type of input fields. We search for matches in the corresponding <label> tag and visible text, and in attributes such as autocomplete, type, label, placeholder, and value.

Once we determine the input type, we check which input fields must be filled as indicated by the presence of the "required" attribute, an '*,' or a bold label. Then we fill all the required inputs by simulating typing. We ensured that our fictitious credentials including an EU address seem plausible. This, together with VPN in the EU, should indicate for the website that EU privacy laws are

applicable, which we further discuss in Section 7. Most importantly, we generate a unique email address for every website.

*Checkboxes and form submission.* We interact with every required checkbox and <select> tag. Once the form is filled, we submit it using any detected submission button or by simulating pressing the Enter key. After submission, we look for a redirect or a change in the website content to detect the registration state. We compute the difference in the website's visible content and the form code to distinguish the following outcomes. The text differs and contains keywords indicating a 'successful' or 'failed' registration. The form remains unchanged, usually indicating a 'failed' registration. The form is changed after a redirect, indicating a multi-step registration. None of the above applies and we denote this an 'unknown' state.

If the registration failed but the same form is still present, we try filling in the credentials again, but this time we confirm all checkboxes. This increases the probability that a required checkbox like "I agree with the terms and conditions" is checked. However, it also increases the probability of consenting to sending marketing emails, which could be detrimental to the objective of our consent study.[4] Then the form is submitted again, possibly many times when the form changes and our heuristic detects a multi-step registration.

*CAPTCHA solving.* During any of the crawling steps, we might encounter a CAPTCHA. This usually happens during registration or when loading an index page is intercepted by CloudFlare or a similar DDoS-mitigation service. The crawler observes the type of CAPTCHA by the JavaScript that loads it. For reCAPTCHA or hCAPTCHA, we load a template substitute JavaScript that prevents crashes due to website changes of the CAPTCHA invocation. Image CAPTCHAs are detected by keywords directly in the forms. We use an external service that solves CAPTCHA using humans.

A third of crawled websites use CAPTCHAs: 75% of them ReCaptcha v2, 20% ReCaptcha v3, 2% hCaptcha, and 3% image CAPTCHA.

*Self-hosted mailserver.* We self-host generated email addresses at *redacted for review*, configured to only receive emails using the Mail Delivery Agent implemented with the Python Maildir library.

## 3.2 Registration confirmation

Once the crawler determines that the registration state is either 'successful' or 'unknown,' it waits for a confirmation email. As shown in [21], only 85% of websites send emails to registered users and, of those, 59% send double-opt-in emails requiring activation. If we receive an activation email, we extract the activation link or code. The crawler visits the activation link or inserts the code into the open registration.

For computational reasons, we wait for activation mail only for a limited period. We discuss this period and issues that we faced with confirmations in Appendix A.4.

## 3.3 Deployment

We evaluated our crawler by visiting the Tranco 1M list[5] [23], generated on 15 June 2022. We selected the Tranco list to enable an

---

[2]TLD+1 refers to the registered domain name preceding the top-level domain. For example, in both bbc.co.uk and bbc.com, the string 'bbc' represents the TLD+1.
[3]From an older crawl using https://tranco-list.eu/list/89WV/1000000.

[4]Checking all checkboxes hinders detecting the 'marketing email despite user did not consent' violations. Skipping such a requirement improves the registration rate by 10%, which is relevant for our crawler application to other than consent compliance studies.
[5]Available at https://tranco-list.eu/list/82Q3V

accurate comparison with prior work that utilizes a similar crawling list. However, Ruth et al. [30] have observed that Tranco represents less accurately users' browsing patterns than the Chrome UX Report (*CrUX*) list. Hence we also evaluate the subset of Tranco that is present in the CrUX list. Unfortunately, due to a processing error, we crawled one million websites that were uniformly randomly sampled with replacement, rather than crawling all the websites. For this reason, our results are only based on 660 202 unique domains, corresponding to the first crawl.

The crawl was conducted from June to September 2022, averaging 10k websites per day on a server equipped with four Intel Xeon E7-8870 CPUs. We ran 60 Chrome browsers in parallel each within a separate docker container, using a freshly launched browser for every website. We used 16 IP addresses provided by the German Research Network ensuring that the traffic originates in the EU.

The crawler collected evidence in the form of HTML code from the index and registration pages, as well as extracted text from the privacy policy and terms and conditions. Additionally, we obtained screenshots of each step taken during registration and recorded all the observed cookies. Finally, the crawler collected information regarding the registration status, which we describe below.

### 3.4 Crawling results

From the 660 202 websites, 504 509 websites were successfully loaded in a supported language. Among the loaded websites, our crawler detected a `registration` or `subscription` form on 33.6% (169 765) of them. Furthermore, our crawler estimated the success rate of form submissions defined in Section 3.1.3. The estimation indicates that 30.2% of form interactions were successful (51 290), 38.4% failed (65 220), and 31.4% resulted in an undefined state (53 255).

The form submissions state detection is prone to false positives. Hence we manually investigated the correctness of the crawler determined registration state by inspecting 200 websites and testing the used credentials. The analysis revealed three newsletter subscriptions deemed successful by the crawler and nine registrations, seven of which were correctly identified as successful by the crawler. Two registrations were successful, despite the crawler assigning them an 'unknown' and 'failed' state. We suspect that newsletter forms were underrepresented in this sample and as nearly half of the received emails resulted from newsletter subscriptions. Further observations from the manual analysis are presented in Appendix B.

We also analyze the results based on whether the websites are included in the CrUX list. Note that Tranco 1M and CrUX have only 51.9% overlap. The crawl was significantly more successful for the CrUX websites. Specifically, 90.6% of the websites present in both lists were successfully loaded, in contrast with 65.3% for non-CrUX websites. Among the websites in the CrUX list, registration was detected as successful in 11.7% of cases (3.9% for non-CrUX websites). Our list choice supports a comparison with [11], relying on the DNS-based Alexa list with domains as `WindowsUpdate.com` without HTTP(S) endpoint. In the future, we recommend crawling the CrUX list to prevent unnecessary computations.

### 3.5 Ethical considerations

We have identified the following three risks of our study. 1) *Legal risks arising from crawling*: we considered various legal regimes and concluded that our research does not violate laws such as fraud, trespass, or breach of contract as our intentions are the pursuit of good-faith privacy research. 2) *Risks to website owners*: our single crawl negligibly impacts each individual website's capacity. Moreover, the registration rarely results in a manual action by website owners, as the vast majority of emails are automated. In Section 5, we present only aggregated results, preventing harm by wrongful accusation of individual websites for privacy violations. For that reason, we refrain from publicly disclosing our dataset of identified violations, except in cases where parties explicitly provide consent to adhere to the same ethical standards we uphold. 3) *Risks to CAPTCHA solvers*: we employed a third-party CAPTCHA solving service. Given the substantial prevalence of CAPTCHAs, accounting for one-third of our successful registration, and their prevalence on often higher-profit services, omitting CAPTCHA solving would introduce a significant bias. Finally, we discussed the outsourcing with our university's legal department. Furthermore, our university's ethics board determined that our project does not require ethics approval as it does not involve human subjects. Since email service providers have started to require CAPTCHAs to complete the confirmation, we are transitioning to CAPTCHA solving by research assistants employed at our university.

## 4 CLASSIFYING LEGAL PROPERTIES

Kubicek et al. [21] defined 21 legal properties relevant to consent compliance and annotated a dataset with them. In this section, we automate the prediction of these properties. Using the dataset from [21], we train two types of ML models: for emails and forms. For each type of model, we describe the feature engineering step, how models are trained, and the results.

### 4.1 Features of emails

The training dataset consists of 5725 mostly German and English emails. To reduce the complexity of dealing with multiple languages and to utilize all the training samples, we translate the subjects and bodies into English using LibreTranslate. From each translated email, we further process the headers, subject, and body.

*4.1.1 Headers.* Email headers constitute a set of key-value string pairs, such as 'Date,' 'Reply-To,' or 'List-Unsubscribe.' While several headers are standardized, there are many, often prefixed with 'X-,' that are custom to specific email servers. We define the *supported keys* as the set of all header keys in the training dataset. This resulted in 76 headers without the 'X-' prefix and 488 headers with it. For each email, we denote whether there is an entry for a given key, whether it contains an email, URL, other text, or whether it is empty.

*4.1.2 Subject.* The translated subjects are processed with TF-IDF encoding[6] that we fit to the training dataset. In addition to this encoding, we use a universal sentence encoder [5]. This pretrained NLP model transforms sentences to an embedding in $\mathbb{R}^{512}$.

*4.1.3 Body.* We extract both the TF-IDF encoding of the translated body and several manually-defined numeric features. These features

---

[6]Term Frequency-Inverse Document Frequency (*TF-IDF*) is a variant of the Bag-of-Words text representation model that accounts for the total number of words. It outperforms Bag of Words in common classification tasks [1].

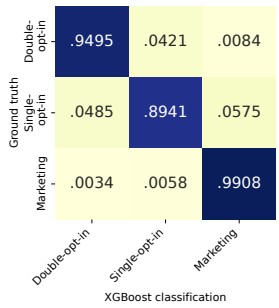
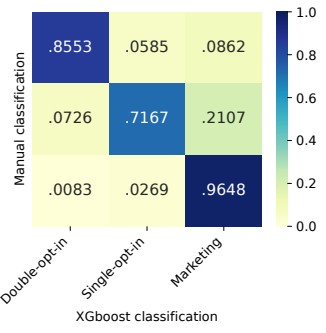

**(a) Using training dataset by Kubicek et al. [21].**

**(b) Using our labeled 11k emails.**

**Figure 2: Confusion matrices of mail type classification.**

include the number of characters or sentences of the email text, number of URLs, images, and links.

### 4.2 Training ML models for emails

Given that our features correspond to tabular data, we use the XGBoost model [7]. XGBoost is well-suited as it outperforms other training algorithms for datasets with few annotated samples but high dimensionality of the feature space.

We train the model using an established ML pipeline. We perform a stratified split of the dataset dedicating 75% for training, saving 25% of the unseen data for validation. We adjust for class-imbalance by sample-weighting. The models optimize the weighted 'multi:softmax' metric for multi-class classification and 'binary:logistic' for binary classification. All reported results are based on four-fold cross-validation. Given data scarcity, we skip hyperparameter tuning, which would require a further data split, and we use the default XGBoost hyperparameters.

We trained models that predict two distinct legal properties of emails. Our first model predicts whether an email is a marketing email (i.e., newsletters, notifications promoting service monetization, and surveys), a servicing double-opt-in email, or another kind of servicing email (confirmation emails or service updates). Our second model detects whether an email contains a method to unsubscribe, which we evaluate only on marketing emails.

In Fig. 2a, we present the confusion matrices of the mail-type model. The mail-type model achieves 97.7% balanced accuracy, while in the simplified task of deciding only whether email is marketing or servicing (aggregating double-opt-ins with confirmations and legal updates), the balanced accuracy increases to 99.2%. The same balanced accuracy of 99.2% is achieved by the model predicting the presence of the unsubscribe options.

### 4.3 Features of forms

To transform forms of unlimited length to tabular features, we aggregate the form inputs by the crawler's keyword-based element classification. We group semantically similar inputs, such as the first and last name, full name, and username, see Appendix A.3 for details. We also reduce the complexity by excluding inputs irrelevant to legal classification, such as CAPTCHA. From all inputs, we extract whether they are required or optional, and from checkboxes also

their default values. We concatenate texts, such as corresponding labels, and translate them to English. Finally, we include the form type (registration or subscription) as a categorical feature.

We then process the form texts similarly as emails. Note that checkbox labels often consist of complex and nuanced statements, such as "I don't want to receive special offers about [company name] products." To better capture the meaning of these statements, we extract both sentence embeddings and TF-IDF representations with a limit of 500 words. However, for other form inputs, which tend to have shorter labels like "Your email," we skip sentence embeddings and only use TF-IDF with a limit of 50 words.

The feature extraction produces 5839 tabular features: 69 numerical features about form's input fields, 3154 TF-IDF columns, and five sequences of $\mathbb{R}^{512}$ sentence embeddings.

### 4.4 Training ML models for forms

Similarly, as with the email classification, we trained an XGBoost model for each of the 21 binary legal properties annotated by [21]. Note that the training dataset consists of only 668 annotated forms. To address this data scarcity, we also conducted experiments using the Tabnet model [2], a neural network model optimized for tabular data. One notable advantage of Tabnet over XGBoost is its ability to perform unsupervised pretraining on unlabeled data, allowing it to capture the distribution of classified data. For the pretraining phase, we provided the extracted features of 30k websites where the crawler detected registration or subscription forms.

Table 1 presents the results of XGBoost with predictions based solely on the crawler's keyword-based classification of form content. However, the crawler's prediction is unavailable for some legal properties, so for space reasons we skip such rows together with Tabnet as its performance is aligned with that of XGBoost. The table provides a summary of the macro-averaged F1 score, precision, and recall, while the last column indicates the percentage of positive samples in the training dataset. Note that the overall performance is highly dependent on the number of positive samples, making scarce properties insufficient for making legal judgments. To mitigate the risk of falsely predicting a privacy violation, we combine the ML predictions with the crawler's keyword-based deterministic prediction. When the presence of a legal property implies a violation, we combine predictions using AND and conversely when it implies compliance, we use OR. We further reduce false positives by conditioning predictions when possible, such as 'marketing checkbox forced' requires 'marketing checkbox present' in the first place.

## 5 POTENTIAL VIOLATION DETECTION

In this section, we describe our analysis of security threats and potential privacy violations concerning consent in forms and emails. For each method, we give context regarding related work and the EU privacy regulations – the GDPR and the ePrivacy Directive.

### 5.1 Security violations

Using our automated methods, we investigate websites' adherence to security best practices in private data protection mandated by Art. 25 and 32 GDPR. We focus on the collection of personal information through user registration and newsletter sign-up processes.

**Table 1: Performance of legal properties models. 'Determ.' model stands for the crawler's prediction.**

| Property | Model | F1 | Precision | Recall | Support |
|---|---|---|---|---|---|
| Marketing consent | Determ. | 77.58% | 80.43% | 76.87% | 41.92% |
| | XGBoost | 82.33% | 82.88% | 82.08% | |
| Marketing purpose | Determ. | 68.06% | 64.95% | 74.65% | 7.04% |
| | XGBoost | 63.15% | 61.71% | 66.21% | |
| Marketing checkbox present | Determ. | 79.01% | 83.23% | 77.33% | 35.18% |
| | XGBoost | 81.67% | 82.95% | 81.04% | |
| Marketing checkbox pre-checked | Determ. | 71.74% | 73.26% | 70.44% | 5.84% |
| | XGBoost | 57.66% | 57.58% | 58.43% | |
| Marketing checkbox forced | Determ. | 55.67% | 59.67% | 54.22% | 3.14% |
| | XGBoost | 58.94% | 59.84% | 58.38% | |
| Tying policy and terms checkboxes | Determ. | 71.16% | 71.48% | 70.86% | 16.77% |
| | XGBoost | 77.71% | 78.10% | 77.92% | |
| Tying all checkboxes | Determ. | 51.84% | 51.51% | 64.49% | 0.45% |
| | XGBoost | 49.70% | 49.70% | 49.70% | |
| Forced policy | XGBoost | 74.16% | 74.34% | 74.07% | 26.95% |
| Forced terms | XGBoost | 74.05% | 80.28% | 70.99% | 5.24% |
| Forced policy and terms | XGBoost | 72.55% | 72.16% | 73.25% | 18.41% |

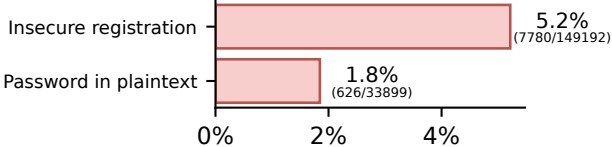

**Figure 3: Security threats of registration to websites.**

We present our findings in Fig. 3. We detect 5.2% of websites collecting sensitive information through forms from unsecured HTTP websites, failing to ensure the personal data confidentiality required by GDPR. Utz et al. [34] found such violation on only 2.85% of websites. The difference might stem from our better selection of forms for inspection and the difference in the crawling lists. We also observed 1.8% of websites that send us an email included the user-provided password in plaintext in the email. The data protection authority of Baden-Württemberg (Germany) [3] considers this practice as a violation of Article 32 of the GDPR. A similar incidence of 2.3% was observed in manual study of Kubicek et al. [21].

## 5.2 Violations of marketing consent in forms

Our detection of potential violations of marketing consent in forms is based on the predicted legal properties used in the decision procedures defined by Kubicek et al. [21, Figs. 6 and 7]. Due to space constraints, in Fig. 4 we only report the aggregated results using these procedures. Note that the baseline of reported incidence is 33 899 of websites that send any email. According to Kubicek et al. [21], only 85% of registrations result in the website sending any email, and this factor should be taken into account when interpreting our results.

Over 43% of registrations resulted in websites that never sent us any marketing emails, potentially caused by issues with account activation (see Appendix A.4) and up to 44% of the marketing emails we received resulted from newsletter subscriptions, reflecting the crawler's higher success rate with `subscription` forms compared to `registration` forms. We found that at least 3.6% of senders violated the opt-in requirement of the ePrivacy Directive by sending marketing emails without any indication of marketing email

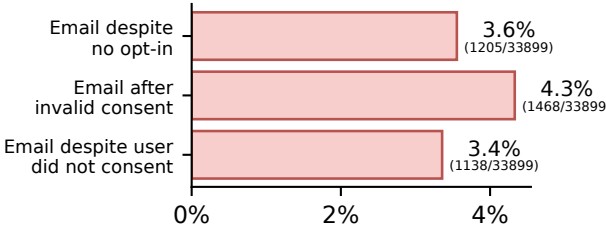

**Figure 4: Portion of senders that violate at least one marketing consent requirement. This figure is based on the decision procedures from [21, Figs. 6 and 7].**

consent. At least 4.3% of websites then violate the GDPR consent requirements by not including a marketing checkbox, pre-checking the checkbox by default, or tying the checkbox with privacy policy or terms. In 3.4% cases, we received a marketing email despite rejecting consent, where the checkbox was neither pre-checked nor checked by the crawler.

## 5.3 Email privacy violations

When users register, websites should verify the ownership of the registered email address through a double-opt-in process. Without this verification, our crawler could be used to subscribe arbitrarily selected email addresses to thousands of newsletters without the owners' consent, resulting in the Bomb attack [31]. The double-opt-in process also ensures that the website retains a clear record of consent. Using the ML model from Section 4.2, we classify the first email we receive from the website. The results presented in Fig. 5a show that 42.4% of websites adhere to the double-opt-in requirements and 24.8% of websites only send a confirmation email, not conforming to the double-opt-in practice. The remaining 32.8% of websites immediately send marketing emails to users.

*5.3.1 Email sharing.* To track how websites use email addresses, each registration was performed with a unique email address. Detecting when the website shares the email address to third parties, however, poses a challenge. For example, facebook.com sends emails from facebookmail.com. We developed the following heuristic to address this issue.

For a given registration, we extract a set of TLD+1 domains from which we receive emails. We then match these domains to other domains found in various sources documenting how the website declares this domain. We consider that domains match if the longest common subsequence between two domains is at least half of the shorter domain. This threshold of 0.5 was determined by empirical evaluation of a set of 200 domain matches, resulting in an accuracy of 91% with 2.5% of false negatives (wrongly predicting that domains are not similar) and 7.5% of false positives.

For each sender domain, we identify how the website discloses it. We take the first of the following outcomes, ordered from the most to the least disclosed. (1) The domain name where we registered and any domains that are similar are marked as 'registration domain.' (2) The domain of the first received email is marked as 'first sender.' (3) Any common email host (e.g., gmail.com) is marked as 'similar

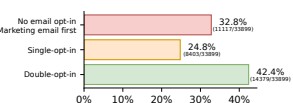

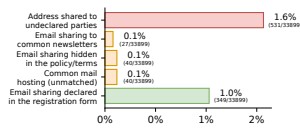

**(a) Email opt-in classification of the first received email.**

**(b) The detected types of email sharing.**

**Figure 5: Potential violations in emails.**

email host' if the name preceding the @ symbol is similar to the registration domain. (4) Any domain declared on the registration page is marked as 'in form.' (5) Any common host that was not matched previously as 'dissimilar email host.' (6) Domains in the privacy policy and terms and conditions, are marked as 'in policy/terms.' (7) Domains that belong to common newsletter senders such as Mailchimp are marked as 'newsletter sender.' (8) If all these checks fail, the domain is marked as 'undeclared.' We list other methods we considered for third-party sharing detection in Appendix C.1.

If there are at least two senders and one of them is marked as 'dissimilar email host' or higher in the ordering above, we consider the website to be sharing the email address without a proper disclosure. As shown in Fig. 5b, 1.6% of our email addresses received emails from undeclared domains, including one website that shared our email address to 56 undeclared domains. Additionally, 0.3% of websites sent emails through common newsletter senders such as MailChimp or from domains that were only declared in the policy or terms, which are rarely read [4]. Finally, 1.0% of senders are correctly defined directly in the form, and the remaining websites sending emails do so from expected domains. The prevalence of this violation is comparable to results by Kubicek et al. [21].

## 6 MANUAL EVALUATION

To evaluate the trustworthiness of our automated methods in a real-world scenario, we manually analyzed a random sample of 100 websites that had sent us at least one email. We selected this sample for two reasons. First, it maximizes the number of websites for which our crawler has successfully filled out the form. Second, websites that had sent us emails serve as a baseline for reporting violations. Among these 100 websites, our crawler submitted one `contact`, 54 newsletter `subscription`, 45 `registration` forms. Our crawler misclassified six `subscription` forms as `registration` forms and one `registration` and `contact` form as `subscription` forms.

Out of the registrations or newsletter sign-ups, the crawler was unable to complete 25 double-opt-in procedures. Note that our evaluation of failed double-opt-ins is conservative since we classified any lack of email confirmation as a failure, regardless of whether the website actually sends such an email. Nonetheless, considering that almost half of the websites use double-opt-in, email confirmation should be improved in future work. Additionally, two registrations were incomplete, but the websites reminded us to finish the registration—a behavior that was studied by Senol et al. [32]. Finally, the crawler successfully submitted the remaining 73 forms.

We examined the email opt-in violations and found that the first emails from 83 websites were correctly classified. Unfortunately, the model misclassified that the first email was for marketing rather

than single- or double-opt-in in nine and five cases, respectively. For subsequent legal work, we completed double-opt-ins manually, which allowed us to inspect 11k classifications of initial email, which we summarize in Fig. 2b. Our inspection suggests that we tend to classify emails more rarely to be marketing compared to the annotators of the dataset we used for training [21]. As future work, we will incorporate the larger annotated dataset for training to improve the mail-type model's robustness.

Regarding form interface violations, our sample contained 17 marketing consent violations. Our method detected 11 of them, with an 86% accuracy, 82% precision, and 50% recall. The two false positives were misclassification of servicing emails for marketing, but the method correctly identified the form interface problems.

For insecure registration and passwords sent via email, the sample had two violations each, and their prediction was accurate. We expect false positives to occur only if we misclassify a form. We evaluated third-party sharing on 50 websites sending emails from multiple different domains. This sample contained 13 violations. Our method achieved a recall of 85% (two short sender domains were falsely detected on the registration page) and a precision of 79% (three senders used multiple domains belonging to the same company which can be observed only from the email content).

In conclusion, while our results reasonably represent the landscape of violations, individual violations are sometimes incorrect. Therefore, individual violations should not be blindly trusted without inspecting the evidence we collected. Still, using our detection methods as a tool for privacy enforcement can considerably streamline the detection of violations, as it presents enforcement agencies with a set of potential violations alongside the evidence needed to manually check whether the violation actually took place.

## 7 LIMITATIONS

*Bias.* Our study is susceptible to a selection bias introduced by the crawler. As explained in Section 6, our crawler exhibits greater success in signing up for simple websites and forms such as newsletters compared to complex registrations. However, it is possible that form complexity and website compliance are correlated. Hence, our results may not be representative of the entire population of websites visited by users.

To mitigate this limitation, we propose involving real users in part of the process. For example, semi-automated techniques can be employed for email confirmation, ensuring that humans accurately handle the various double-opt-in processes used by websites. Additionally, violation detection can be similarly inspected.

*Accuracy.* All of our findings are prone to misclassification. Hence all violations should be regarded as potential violations. In particular, in cases where our methods exhibit low precision in identifying violations, caution should be exercised when using the results for enforcement purposes. We propose two complementary solutions to address this. First, one can carefully examine the evidence of the violation in the form of screenshots and website source code similarly to our approach in Section 6. Moreover, a larger training dataset can be constructed by rectifying misclassified violations and adjusting the corresponding legal labels, thereby improving our models in the future. This is particularly crucial for properties with few positive samples, such as the pre-checked marketing checkbox.

Finally, our methods are not a complete audit as there may be additional unaddressed violations. Detecting email sharing might require a longer observation period to capture incriminating events.

*Adversarial websites.* Website operators could modify their forms, for example by including input fields or text labels invisible to users, to evade our violation detection methods, as was proposed by Zhao et al. [36, 37]. We assume that websites do not do this, since we have not published our violation detection models, making it difficult for websites to exploit their weaknesses to evade detection. Moreover, the classification also depends on crawler's keyword-based prediction.

*Territorial applicability of EU privacy laws.* Although we access the websites from Germany and register a user located in the same country, note that websites with only a few EU visitors may not be obligated to comply with EU regulations. To ensure the enforcement of EU law, future studies can restrict their analysis to lists that are ranking websites by the origin of visitors, such as CrUX or Similarweb. In Section 3.4, we found that the registration rate is favorable when crawling such lists. By utilizing these lists and considering additional factors such as the website's language, one can estimate whether a website is targeting users located in the EU and, consequently, whether their privacy rights must be respected.

## 8 RELATED WORK

Drakonakis et al. [11] automated the registration process to detect insecurely configured cookies on over half of the websites. Their crawler registered successfully on 1.6% of Alexa top 1M websites, while our crawler achieved registrations on 5.9% of websites from the comparable Tranco list, although nearly half of our registrations can be attributed to newsletter sign-ups. In contrast, Drakonakis et al.'s method also relies on Single Sign-On (SSO) as part of their procedure, which is unsuitable for our mail violation detection requiring a unique email address for each registration. We attempted to re-evaluate their results, without success as their code is dependent on an outdated Google's SSO API. Zhou et al. [38] registers and inspect vulnerabilities specifically on websites with the Facebook SSO, making their work even less aligned with our study objectives.

A similar crawler was proposed by Chatzimpyrros et al. [6]. They claim that their crawler successfully registered on 26.4% of websites, which accounts for 80% of websites with any form. However, their claims are questionable. First, they regard `login` as `registration` forms. Second, they consider form submission as a successful registration. Finally, they do not report the number of senders, except for 0.03% of websites sending emails without crawler's form submission. Senol et al. [32] similarly investigated the detection of private data exfiltration prior to form submission. They found that nearly 3% of websites extract private inputs, such as email addresses.

Jonker et al. [18] developed a crawler that logs into websites using a legitimate crowd-sourced database of credentials called Bug-MeNot. They were able to login to 14.3% of approximately 50k websites present in the BugMeNot database, but they do not present any privacy or security results. While Jonker et al.'s approach is more effective in logging-in than our crawler, it is limited by the size of the BugMeNot database. Consequently, their approach is unsuitable for detecting violations during the registration process or in emails.

Englehardt et al. [12] automated newsletter subscription, which was successful on 5.7% of US e-commerce websites. They focused on identifying the presence of email tracking and email sharing, revealing third-party sharing by 30% of websites. Mathur et al. [26] studied the 2020 US political campaign with similar observations. In contrast, our research uncovered email address sharing by only 5.2% of the senders. This discrepancy suggests that privacy regulations such as the GDPR foster the protection of privacy, particularly in contrast to jurisdictions that lack similar regulations. Additionally, our crawler was more successful in subscribing to newsletters compared to these works.

Oh et al. [29] studied how website forms meet the GDPR consent requirements, specifying four conditions on consent with privacy policies and terms, including consent presence and tying of checkboxes. We focus on consent to marketing emails, and our methods involve observing the actual data use that violates the consent. Hasan Mansur et al. [15] automated the dark pattern detection across websites and apps, including the identification of pre-checked boxes as a default choice. Their findings however underscore the difficulty of detecting this type of violation. A comparable yet manual study was carried out by Gunawan et al. [14].

Consent compliance was thoroughly studied for subpages of websites that do not require prior user authentication. The focus of researchers lay mostly on cookie pop-ups and the consented privacy policies. We refer to a meta-study by Kretschmer et al. [20] that lists and compares publications with these two focal points.

## 9 CONCLUSIONS AND FUTURE WORK

We have developed a crawler capable of conducting large-scale studies on the privacy and security of website registration. Our crawler more than doubles successful registrations of prior work; signing up to 5.9% of 660k websites. This led to the collection of over 2 million emails. Using this crawler, we were able to detect a wide range of privacy and security threats, fully automating previous manual studies and scaling them by orders of magnitude. To do so, we automated prediction of complex legal properties of forms and emails using ML. We observed 12 605 websites, which is 37.2% of the websites sending us emails, containing at least one potential violation, or sending a marketing email as the first email.

Our automation fosters various kinds of research. First, our crawler enables future work to analyze the privacy and security of authenticated sections, reflecting how real users browse websites. Second, the option to collect a large-scale dataset of emails can foster the research of communication practices. Examples include analyzing whether websites respect the unsubscribe action or studying whether tracking by third-parties is even more present in those parts of websites requiring authentication.

In future work, we will explore using our infrastructure for regulatory enforcement. Namely, by extending our training datasets, such as the annotation of 11k emails, we plan to enhance the predictive capabilities of our machine learning models in detecting violations. These enhanced methods can potentially help understaffed and under-resourced data protection authorities by pre-filtering non-compliant websites and collecting supporting evidence. This can support efficient enforcement at scale and thereby improve security and privacy for users of the web.

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

# A  CRAWLER

## A.1  Bot-evasion techniques

We implemented the following methods to further decrease the chance of our crawling being detected as a bot activity.

*Browser.* We use Undetected Chromedriver,[7] which extends the usual Chromedriver with numerous bot evasion techniques, such as removing fingerprints unique to Selenium. Unfortunately, there is no equivalent driver available for Firefox.

*Fingerprinting evasion.* For each page load, the crawler checks the load status. This functionality is not directly implemented by Selenium, so we use Chrome DevTools Protocol for Chrome and Selenium Wire for Firefox. The use of Selenium Wire is however prone to TLS fingerprinting. The proxy and browser differ in the ciphersuite, which is inspected by modern bot detection systems like Cloudflare. While the Firefox-based crawler is prone to this detection, the Chrome implementation does not use any proxy. Additionally we must run Chrome with a non-root user. Chrome disables sandboxing protections when run as root, making it flagged as a bot by Cloudflare.

*Interaction speeds.* Interactions with the website cannot occur instantaneously, as humans are limited in their reading and writing speeds. Our crawler introduces random time delays before each click and during typing to mimic human behavior.

*IP address.* As we study the impact of the EU's privacy regulations, we focused our data collection on traffic originating from within the EU. We considered using commercial VPNs, datacenter or residential proxies, or a university VPN located in the EU. According to a study by Demir et al. [9], residential proxies are the least likely to be detected as bot traffic, closely followed by university VPNs, while datacenters and commercial VPNs are blocked more frequently. Since purchasing a large number of residential IP addresses from services like Bright Data is expensive (≥$10k for our crawl), we used a VPN provided by a university in Germany, which gave us access to a block of 16 IP addresses.

## A.2  Supported languages

Our crawler supports 36 languages, with most of the keywords being translated by native or fluent speakers of the language, whom we instructed in collecting multiple example websites prior to the translation. These languages are: Bulgarian, Bosnian, Catalan, **Czech**, Welsh, **Danish**, **German**, **Greek**, **English**, **Spanish**, Estonian, Basque, **Finnish**, **French**, Galician, Croatian, **Hungarian**, Icelandic, **Italian**, Luxembourgish, Lithuanian, Latvian, Macedonian, Maltese, **Dutch**, Norwegian, **Polish**, **Portuguese**, Romanian, **Russian**, **Slovak**, Slovenian, Albanian, Serbian, **Swedish**, **Turkish**, and **Ukrainian**. From these languages, only 18 of them are supported by LibreTranslate and therefore are suitable for detection of all the violations. We highlighted these languages in bold.

## A.3  Crawler form classification

Our crawler distinguishes various form fields, which we aggregate to the following groups for the form feature processing. This fixed

structure allows us to process differently ordered forms using the same tabular pattern.

- mail
- password
- phone
- username
- names: first, middle, last or full name
- name-other: organization, title, honorific prefix, other text fields
- address: street, house number, city, ZIP, country, full address
- age
- sex
- checkbox: terms of service
- checkbox: privacy policy
- checkbox: privacy policy and terms of service
- checkbox: marketing, privacy policy and terms of service
- checkbox: marketing
- checkbox: SMS
- checkbox: age
- checkbox: other
- birthday: day, month, year, full birth, other <select>
- submit buttons: registration, subscribe
- other buttons: login, contact, other

## A.4  Email confirmation

Since letting the crawler wait for an activation email is computationally expensive, our crawler only waits for up to 30 seconds. If an activation email is received after this period, we activate the registration using a standalone script that processes the incoming emails from all the crawlers running in parallel. However, this script lacks the registration page session, such as cookies, which reduces its success rate compared to the stateful crawler within the 30-second period. We analyzed the distribution of confirmation emails over time in our crawl and observed that less than half of the activation emails arrived within this 30-second period. To achieve a higher success rate for account activation, we recommend waiting for five minutes in future work, since 97.7% of websites that send activation emails do so within this period. Further increasing the waiting period to, say, fifteen minutes would only marginally improve this rate to 99.0%. The longer waiting time, however, comes at the expense of crawling time. Specifically, waiting for five minutes doubles the crawling time, while waiting for fifteen minutes almost quadruples it.

Unfortunately, due to technical issues the independent confirmation script was malfunctioning for about half of the crawl. The combination of a shorter period of waiting by the crawler and the faulty script results in lower confirmation rates. This causes the presented results in Section 5 to be more conservative. Namely, websites that violated the consent in the form but then complied with the double-opt-in requirement and never sent us a marketing email are falsely considered compliant.

# B  MANUAL ANALYSIS OF THE CRAWLER

We conducted a manual investigation of 200 crawled websites to evaluate form detection. Out of the 200 pages, 19 failed to load, and

---

[7]https://github.com/ultrafunkamsterdam/undetected-chromedriver

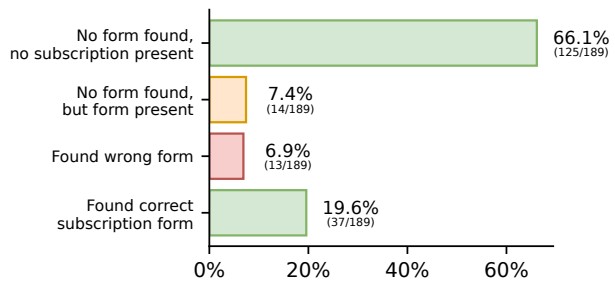

**Figure 7: Evaluation of crawler-detected `registration` forms.**

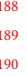

**Figure 8: Evaluation of crawler-detected `subscription` forms.**

thus the analysis presented below pertains to the remaining 181 websites.

In Fig. 7, we present the evaluation of `registration` form detection. Among the sampled websites, 55 had a `registration` form, of which our crawler successfully detected two-thirds. Additionally, the crawler identified a wrong form (e.g., a `contact` form or password reset form) in 10.5% of the evaluated websites. Furthermore, in 4.7% of the websites, the crawler misclassified a `subscription` form as a `registration` form.

Fig. 8 illustrates the evaluation of discovered `subscription` forms. Our findings reveal that 73.0% of websites do not have a `subscription` form (although note that many websites contain both a `subscription` form and a `registration` form). The crawler accurately determined the absence of this form on two-thirds of the websites, and on 19.6% of the websites, it correctly identified the existing form. However, the crawler failed to detect the `subscription` form on 7.4% of the analyzed websites, and in 6.9% of websites, it found an incorrect form.

We also inspected the detected privacy policies and terms and conditions on a list of 300 websites. Our manual evaluation showed that almost 80% and 70% of websites contain privacy policies and terms and conditions, respectively. Our crawler can then detect the correct privacy policy on 51% of websites and correctly conclude that there is no policy on 21% of websites. On 19% of websites, it fails to find the policy and in the remaining 9% of cases, it finds a wrong document. The crawler is correct in finding the terms and detects the absence of terms on 37% and 21% of websites, respectively. It

failed to detect terms on 13% of websites and in the remaining 29% of cases, it detects a wrong document.

## C VIOLATION DETECTION

### C.1 Alternatives for detecting 3rd-party sharing

In addition to the described methods in Section 5.3.1, we explored the following methodologies to minimize false positives and negatives in our violation detection for third-party sharing.

*TLS certificates.* We considered the extraction of company information from TLS certificates. However, note that only a minority, less than 30% of websites, include company names within their TLS certificates. This practice is predominantly observed among highly popular websites, whereas our automated crawling and classification methods perform the best on websites of medium popularity. Furthermore, our observations revealed that websites associated with the same parent companies commonly employ different company names in their certificates, dismissing the usefulness of this approach.

*Co-occurrences.* We investigated the co-occurrence of senders who send emails to multiple addresses registered by our crawler. This analysis uncovered two distinct scenarios. First, email hosting providers such as Gmail were observed to send emails to multiple accounts, suggesting that co-occurrence could be indicative of websites that are compliant with privacy regulations. Conversely, we identified clusters of websites that shared email addresses among themselves without belonging to the same corporate group and without obtaining proper user consent, which strongly indicated privacy violations.

*Company databases.* We explored the use of databases such as Whois, Crunchbase, and Orbis to discover connections between domains owned by the same companies. However, Whois data has become increasingly sparse due to privacy concerns. Moreover, both Crunchbase and Orbis feature inconsistent company name records, leading to false positive violation reports and occasionally attributing incorrect company names, resulting in false negative violation reports. We also considered the webXray dataset curated by Libert [24],[8] but it primarily targets third parties within the tracking industry, which seldom overlap with email senders.

Received 6 October 2023

---

[8]https://github.com/agilemobiledev/webXray/blob/master/webxray/resources/org_domains/org_domains.json

