# OpenReview forum: "Automating Website Registration for Studying GDPR Compliance"
_ACM.org/TheWebConf/2024/Conference — TheWebConf24_

### Official Review · Reviewer_w4E2 · 2023-11-23

**Novelty:** 4
**Technical Quality:** 6

**Review:**

***Paper summary:***

The paper introduces a crawler that automates registrations and newsletter subscriptions, extending automated measurements beyond non-authenticated websites. The authors analyzed 660,000 websites, identifying security and privacy threats within the EU regulatory framework. Additionally, they employed machine learning to assess marketing consent compliance. Findings reveal significant issues, including 37.2% of websites sending marketing emails without proper consent and 1.8% sharing email addresses with third parties without transparent disclosure. The research sheds light on compliance violations and emphasizes the urgency for enhanced data protection measures.



***Detailed comments for authors***

Reasons to Accept the Paper:

+ The paper is well-written and effectively organized, contributing to its overall readability.

+ The authors development of a crawler that automates user registration is a notable contribution. Their effective implementation of techniques to evade bot detection and handle multiple languages resulted in a high success rate, with the crawler successfully loading 90.6% of the targeted websites.

+ The analysis of security threats and potential privacy violations, specifically focusing on consent in forms and emails, showcases the paper's substantial contribution to the field. The authors demonstrate their ability to detect a broad spectrum of privacy and security threats. The automated approach they employed enables the scaling of research efforts that were previously manually intensive.


Reasons Not to Accept the Paper:

 - One limitation that needs consideration is the relatively small size of the testing dataset. With only 426 forms available for page content classification, and an even smaller subset of 163 being registration forms, the authors may want to discuss the potential impact of this constraint on the generalization of their findings.

 - The paper acknowledges a relatively high rate of unsuccessful interactions (38.4%) on websites. To strengthen their research, the authors should delve deeper into the reasons behind these unsuccessful interactions. Understanding the root causes could provide valuable insights and enhance the overall robustness of the study.

**Questions:**

- What limitations are associated with the relatively small size of the testing dataset?
- What is the significance of the relatively high rate of unsuccessful interactions (38.4%) on websites?

**Ethics Review Description:**

The paper effectively addresses ethical concerns.

**Reviewer Confidence:**

3: The reviewer is confident but not certain that the evaluation is correct

**Scope:**

4: The work is relevant to the Web and to the track, and is of broad interest to the community

---

### Official Review · Reviewer_qVZF · 2023-11-24

**Novelty:** 4
**Technical Quality:** 3

**Review:**

The paper presents a comprehensive approach to automating website registration for studying GDPR compliance. The methodology and tools employed are outlined with significant detail, showing a depth of research and application. The paper also discusses the crawler's capabilities, the challenges of bot detection, and the nuances of handling different languages and form types. It provides insights into the technical aspects and the legal implications of GDPR, making it a valuable resource for both technical and legal professionals.
However, I have few concerns:
1.	The approach to translating email content into English using LibreTranslate for ML processing might oversimplify the linguistic complexities. Legal language, particularly in the context of privacy and GDPR compliance, can be nuanced, and this simplification might lead to inaccuracies in detecting violations.
2.	The paper would benefit from an expansion on the error analysis aspect. Including a more in-depth examination of error analysis in the study to enhance the overall completeness and depth of the paper.
3.	While the paper briefly touches upon existing studies, a more comprehensive analysis is needed to clearly articulate the advantages of your approach over them. Specify how your findings align or differ from previous research for a more robust comparison.

**Questions:**

What is the the advantages of your approach over existing studies?

**Reviewer Confidence:**

3: The reviewer is confident but not certain that the evaluation is correct

**Scope:**

4: The work is relevant to the Web and to the track, and is of broad interest to the community

---

### Official Review · Reviewer_v9r8 · 2023-11-25

**Novelty:** 4
**Technical Quality:** 3

**Review:**

*Summary*

This paper develops a crawler to collect large-scale website data by automating website registrations and newsletter subscriptions. The authors find 37.2% of websites send marketing emails without proper user consent. More importantly, they find 1.8% of websites share users’ email addresses with third parties without a transparent disclosure.


*Strengths*

1.This paper studies an important problem. The findings could have broad implications for understanding
2. The proposed method is reasonable and the paper is well-written.

*Weaknesses*

1.The challenges the present study addresses are not clear. The crawlers and email classifier seem quite straightforward. It is difficult to assess the technical depth of the proposed model.
2.The implications of the findings are not sufficiently discussed.
3.The authors do not compare their methods against any baselines.

**Questions:**

Please refer to the weaknesses above.

**Reviewer Confidence:**

3: The reviewer is confident but not certain that the evaluation is correct

**Scope:**

4: The work is relevant to the Web and to the track, and is of broad interest to the community

---

### Official Review · Reviewer_Nu98 · 2023-11-27

**Novelty:** 4
**Technical Quality:** 4

**Review:**

**Summary**
- This research addresses a significant gap in current understandings of online privacy breaches and GDPR violations by creating a web crawler that automates website registrations and newsletter subscriptions.
- The crawler can identify security and privacy threats at scale, offering valuable insights into data privacy and security of websites requiring user authentication.
- The research found that 37.2% of websites send marketing emails without proper user consent, with 1.8% sharing users’ email addresses with third parties without clear disclosure.

**Strong points**
- Proposes a novel crawler tool for automating website registrations and subscriptions, significantly broadening the scope of privacy and security research.
- Leverages machine learning to automatically detect potential violations of marketing consent collection requirements.
- Provides empirical findings around the level of non-compliance to privacy requirements on websites, which is valuable data for policymakers and practitioners.

**Weak points**
- The findings indicate that only 30.2% of form interactions proposed by the crawler are successful. While the authors evaluated the accuracy of form detection through manual analysis, they failed to assess marketing consent and email privacy violation results on the failed domains. It would enhance the authenticity of the project if the authors could manually process a random set of domains and review the results of failed interactions.
- The machine learning model and feature set employed are relatively basic. A potential enhancement could be the use of advanced text embeddings from large language models, instead of currently implemented TF-IDF features.
- There was a processing error during the crawling phase. The authors solely calculated the results based on the 660K unique domains. A further consideration could be to recalibrate the results utilising replacement sampled weights for a better representation of the domain distribution.
- The results could also benefit from additional cross-analysis. Comparisons could be drawn among domain types, domains from different countries, highly populated domains versus others, and more.

**Questions:**

see weak points above

**Ethics Review Description:**

The process reviewed by university’s ethics board

**Reviewer Confidence:**

3: The reviewer is confident but not certain that the evaluation is correct

**Scope:**

3: The work is somewhat relevant to the Web and to the track, and is of narrow interest to a sub-community

---

### Official Review · Reviewer_ZZiB · 2023-11-29

**Novelty:** 5
**Technical Quality:** 5

**Review:**

The paper discusses the development of a crawler to automate website registrations and newsletter subscriptions for investigating how websites handle sensitive user data. This study addresses a gap in automated research, which previously excluded websites requiring user authentication. The crawler is applied to approximately 660,000 websites to identify security and privacy threats in the context of the GDPR and ePrivacy Directive.

The main findings of the paper include the detection of private data collection via insecure HTTP connections and the transmission of user-provided passwords via email. Significantly, the research introduces the application of machine learning to web forms for assessing violations in marketing consent collection. The study revealed that 37.2% of websites send marketing emails without proper user consent, often initiating such emails immediately after subscription. Furthermore, 1.8% of websites were found to share user email addresses with third parties without transparent disclosure​​.

## Strengths
1. **Innovative Approach**: The crawler represents an innovative approach to studying GDPR compliance, addressing a gap in automated research by including websites that require user authentication.

2. **Large Scale Analysis**: The study's large scale, involving approximately 660,000 websites, allows for a wide-ranging assessment of GDPR compliance across various website types and offers a comprehensive overview of the current state of privacy practices online.

3. **Identification of Security and Privacy Threats**: The crawler identifies security and privacy threats, such as private data collection over insecure connections and websites transmitting user passwords via email.

## Areas of Improvements

1. **Lack of Dataset Transparency**: The paper lacks detailed information about the datasets used for evaluating the crawler, especially in relation to Figure 2(b). This omission hinders the assessment of the generalizability and robustness of the findings. More details about how the dataset was curated, how many annotators etc will help the reader understand the dataset better.

2. **Potential for False Positives**: The strategy of agreeing to all checkboxes in case of initial registration failure could lead to false positives, particularly in consenting to marketing emails. This approach might affect the accuracy of the analysis concerning user consent and GDPR compliance.

3. **Absence of End-to-End Evaluation**:The paper does not provide an end-to-end evaluation that accounts for the cumulative effects of various system components. Such an evaluation is crucial to understand the overall effectiveness and limitations of the crawler in practical scenarios.

4. **Use of Classical NLP Techniques**: The authors use TF-IDF to compute the features for classification models. Perhaps the system performance could be improved by using contextual embeddings such as BERT.

**Questions:**

- In cases where the crawler inadvertently agrees to marketing emails as part of its strategy to ensure successful registration, how does this affect the analysis of GDPR compliance regarding user consent? Could you elaborate on the frequency of such occurrences and their impact on your study’s outcomes?

- Could you provide more comprehensive details on the methodologies used, particularly in terms of data processing, model training, and the criteria for evaluating different modules?

- Regarding the dataset specifically mentioned in relation to Figure 2(b), can you provide more information about its curation and characteristics? How was this dataset compiled and what were its key features? A more detailed description of the dataset(s), including their composition and the criteria used for selection, would be beneficial.

- Can you provide details regarding an end-to-end evaluation of the system? An overall system-wide evaluation would allow the reader to understand the effectiveness of the crawler as a whole, providing insights to gauge the crawler's holistic accuracy and reliability.

- Given that the crawler supports multiple European languages, how does the system handle the nuances and cultural variations in terms of privacy policies and consent forms across different languages and regions?

Minor Question:
- How adaptable is your crawler to changes in website designs and user authentication methods? With the dynamic nature of web, how frequently do you anticipate needing to update the crawler?

**Reviewer Confidence:**

3: The reviewer is confident but not certain that the evaluation is correct

**Scope:**

4: The work is relevant to the Web and to the track, and is of broad interest to the community

---

### Decision · Program_Chairs · 2024-01-22

**Decision:**

Accept

**Comment:**

Our decision is to accept. Please see the AC's review below and improve the work considering that and the reviewers' feedback for cemera-ready submission.

"This paper presents a crawler for automating website registrations and newsletter subscriptions to study how websites handle sensitive user data. The crawler is applied to approximately 660K websites to identify security and privacy threats in the context of the GDPR and related directives. The results show than more than 37% of the websites send marketing emails without proper user consent and almost 2% of them shared user email addresses with third parties without transparency.

 Strengths:

 - The crawler represents an innovative approach to study GDPR compliance, addressing a gap in automated research by including websites that require user authentication.
 - The study includes over 600 thousand websites, allowing a large-scale assessment of GDPR compliance that provides a comprehensive overview of current online privacy practices, which is useful for policymakers and practitioners.
 - The crawler identifies security and privacy threats, such as private data collection over insecure connections and websites transmitting user passwords via email, by using machine learning techniques.

 Weaknesses:

 - The datasets used for evaluating the crawler, are not completely well described with respect to its creation, annotation, etc.
 - Only 30% of form interactions proposed by the crawler are successful, so the results are biased to this sample.
 - The ML techniques can be improved as well as the NLP methods.
 - The results could also benefit from additional cross-analysis.

 Scope: 4; Novelty: 4; Quality: 4"